# Lifelong Lung Sequelae of Prematurity

**DOI:** 10.3390/ijerph19095273

**Published:** 2022-04-26

**Authors:** Paola Di Filippo, Giulia Dodi, Francesca Ciarelli, Sabrina Di Pillo, Francesco Chiarelli, Marina Attanasi

**Affiliations:** Pediatric Allergy and Pulmonology Unit, Department of Pediatrics, University of Chieti-Pescara, 66100 Chieti, Italy; giulia.dodi16@gmail.com (G.D.); francescaciarelli89e@gmail.com (F.C.); sabrinadipillo@gmail.com (S.D.P.); chiarelli@unich.it (F.C.); marina_attanasi@hotmail.it (M.A.)

**Keywords:** prematurity, lung function, DLCO, chronic obstructive disease

## Abstract

The clinical, functional, and structural pattern of chronic lung disease of prematurity has changed enormously in last years, mirroring a better perinatal management and an increasing lung immaturity with the survival of increasingly premature infants. Respiratory symptoms and lung function impairment related to prematurity seem to improve over time, but premature birth increases the likelihood of lung function impairment in late childhood, predisposing to chronic obstructive pulmonary disease (COPD). It is mandatory to identify those individuals born premature who are at risk for developing long-term lung disease through a better awareness of physicians, the use of standardized CT imaging scores, and a more comprehensive periodic lung function evaluation. The aim of this narrative review was to provide a systematic approach to lifelong respiratory symptoms, lung function impairment, and lung structural anomalies in order to better understand the specific role of prematurity on lung health.

## 1. Introduction

Since Barker’s hypothesis [1], increasing importance has been given to the effect of early life events on adulthood. New insights suggest that adverse early life events influence long-term health trajectories throughout life.

Worldwide preterm birth (<37 weeks of gestation) affects approximately 10% of live births and is the leading cause of death in children less than 5 years of age [2]. Preterm birth disrupts normal lung development leading to several respiratory complications in the neonatal period and later in life [3]. Although the airways complete at the end of the pseudoglandular stage of fetal lung development (Figure 1), lung parenchyma is immature at birth because the alveolarization starts at 36–37 weeks of gestation and occurs up to early adulthood, mainly between birth and 8 years of age [4]. Consequently, factors that negatively affect prenatal and early life respiratory growth can compromise the achievement of “personal best lung function”. A recent systematic review including 16 studies confirmed the strong association between early life insults and development of chronic obstructive pulmonary disease (COPD) in adulthood. The authors found that prematurity, in utero tobacco exposure, early childhood asthma, and pneumonia increased the likelihood of lung function impairment in late childhood predisposing to COPD [5].

Infants born before 32 weeks of gestation have the greatest risk of mortality and bronchopulmonary dysplasia (BPD). Improved neonatal intensive care has contributed to increased survival of preterm newborns and thus to increased number of children with respiratory morbidities later in life [3].

Acute neonatal respiratory problems, such as respiratory distress syndrome, meconium aspiration, sepsis, persistent pulmonary hypertension, congenital heart disease, and their subsequent treatments, could evolve in chronic lung disease (CLD) of infancy. BPD is the most common form of CLD of infancy associated with premature birth and treatment for respiratory distress syndrome in preterm infants [6].

Northway et al. [7] defined BPD in 1967 as the need for supplemental oxygen at 28 days of postnatal age in preterm infants who required mechanical ventilation for at least 1 week. BPD was characterized by symptoms of persistent respiratory distress and radiolucent areas alternating with radio-dense ones on chest *x*-ray. However, the ‘‘new’’ BPD differs from the “old” BPD described in 1967. The old BPD was mostly caused by medical treatment, particularly by high oxygen concentration and ventilation pressures; anatomically it was characterized by inflammation, airway smooth muscle hypertrophy, emphysema, and parenchymal fibrosis [8]. The new BPD is mostly caused by an extremely immature birth with an interrupted alveolarization, which leads to an impaired alveolar formation, fewer and dysmorphic capillaries, and less evidence of emphysema, fibrosis, and airway changes when compared to old BPD [8].

In 2001, Jobe et al. [9] defined BPD as a persisting oxygen dependency after 28 days from birth and radiographic changes and established severity grading at 36 weeks of post-conceptual age for infants born at gestational ages of less than 32 weeks.

In general, studies on respiratory outcomes after preterm birth have produced inconsistent results. Long-term effects of prematurity on lung function are difficult to investigate because of several methodological problems. Firstly, the heterogeneity of populations and treatments makes difficult the comparison among studies in literature. In addition, recent changes in the medical management of prematurity might have modified the relationship among prematurity, BPD, and lung function over time [10]. Nowadays, the routine use of antenatal corticosteroids and surfactant therapy, and gentler approaches of mechanical ventilation have increased the heterogeneity of the studied populations. Secondly, genetic and environmental factors, such as atopy, tobacco smoke exposure, socio-economic condition, and family history could also predispose to lung function impairment, in addition to prematurity itself [5]. Therefore, this aspect makes difficult to understand both the relationship of these exposures with the increased likelihood of prematurity, and the real contribution of prematurity on respiratory disease onset [6]. To date, most of the studies about prematurity long-term effects on lung function have focused on patients with BPD, while there is increasing evidence of prematurity as a risk factor for respiratory problems even without BPD. In addition, the lack of a homogeneous definition of BPD complicates the comparison of long-term respiratory outcomes between ex-preterm children with BPD and ones without BPD [8]. Hence, the aim of this narrative review was to summarize the state of art about the lifelong effects of prematurity on respiratory diseases, lung function, and structural abnormalities, in order to better understand the specific role of prematurity on lung health.

## 2. Lifelong Symptoms in Ex-Preterm Subjects

In literature, respiratory symptoms and lung function impairment were shown in BPD survivors during childhood and adolescence. Lower levels of exhaled nitric oxide and exhaled breath temperature [6,11] suggested a different mechanism in BPD survivors when compared to asthmatic children [12].

### 2.1. Respiratory Symptoms in Infancy and Preschool Age

During the first years of life, respiratory symptoms were more common in children born very preterm (especially with BPD) than children born at term and, specifically, during the first 2 years of life, preterm infants with BPD suffered more frequently from recurrent [13,14] wheezing when compared to term-born infants [15]. In our previous study, we showed that preschool wheezing was more frequent in ex-preterm children when compared to term-born controls, independently of the presence of BPD [8].

In infancy, ex-preterm children symptoms could overlap with those of rare pathologies, both congenital, such as Mounier Kuhn and Williams-Campbell syndrome, and postinfectious, such as Swyer-James syndrome. Mounier-Kuhn syndrome is characterized by dynamic dilation and collapse during inspiration and exhalation of upper airways, due to a dilated trachea and main bronchi [16]; Williams-Campbell syndrome patients present generalized tracheobronchomalacia due to poor cartilage in the segmental and subsegmental bronchi [17]. Swyer-James syndrome is a postinfectious form of bronchiolitis obliterans and its principal features are decreased pulmonary vascularity and hyperinflation (typically unilateral), with or without bronchiectasis [18].

### 2.2. Respiratory Symptoms in School Age

Respiratory symptom rate decreased substantially in ex-preterm children with BPD during school age, although it was higher than their peers without BPD [13]. In mid-childhood, the association of prematurity with wheezing, shortness of breath, and cough was initially described in the pre-surfactant era [19,20,21,22] and then confirmed in more recent studies [23,24,25,26].

This increased prevalence of respiratory symptoms appears to be independent of BPD [27]. Preterm children with and without BPD reported a two to three fold higher prevalence of wheeze than term-born controls [28]. However, more severe respiratory symptoms were reported in children requiring prolonged ventilation or developing BPD [6].

### 2.3. Respiratory Symptoms in Adolescence

Studies carried out in those of adolescent age reported contrasting results. Doyle et al. [29] found no difference in respiratory health between preterm children and term controls. Contrarily, Anand et al. [30] observed a significantly higher prevalence of chronic cough, wheezing, and asthma in ex-preterm subjects when compared to controls. However, the prevalence of asthma was similar in ex-preterm groups and different in control groups in aforementioned studies, determining a potential bias which could explain these conflicting results.

### 2.4. Respiratory Symptoms in Adulthood

In adulthood, several studies reported more respiratory symptoms in young adults with a history of prematurity. In a prospective cohort study, 60 ex-preterm subjects not treated with surfactant showed more respiratory symptoms when compared to 50 healthy term controls 21 years after preterm birth, although the overall prevalence of respiratory symptoms decreased over time. Interestingly, respiratory symptoms were not necessarily associated to lung function impairment or an abnormal exercise tolerance [31]. Baraldi et al. [32] reported that premature infants with a reduced lung function at birth complained of respiratory symptoms at 18–20 years of age. A higher prevalence of wheezing, pneumonia, and long-term medication use in young adults (mean age, 18.3 years) with a previous history of BPD when compared to term born controls was documented in a multicenter survey [33]. Gough et al. [34] investigated respiratory symptoms and health-related quality of life of ex-preterm subjects, with (72 subjects) and without (57 subjects) BPD, with 78 term born controls at 24–25 years of age. The authors found that BPD survivors had significant respiratory symptoms and quality of life impairment that persisted into adulthood. In addition, the authors reported that 72 BPD subjects showed two-fold higher prevalence of wheeze and three-fold higher use of asthma medication than controls. In 2015 Caskey et al. [35] confirmed that young adult (mean age 24 years old) BPD survivors complained more wheeze, breathlessness, and wakening with cough when compared to non-BPD or term control subjects.

To date, the balance of evidence in adolescence and adulthood suggested that ex-preterm subjects, independent of the presence of BPD, have more respiratory symptoms, such as coughing, wheezing, and asthma, although some studies suggest symptoms may be more frequent in those with BPD [14].

A summary of lifelong respiratory symptoms is provided in Figure 2.

### 2.5. The Exercise Tolerance

Several studies reported a compromised exercise tolerance in children born preterm (both with and without BPD) when compared with term-born controls [33]. A decreased exercise tolerance was indicative of impaired aerobic power, as evidenced by a reduced peak oxygen consumption [25,36,37], less distance traveled on the treadmill [26,35], a greater breathing frequency and a lower tidal volume during peak exercise [25,38] in ex-preterm subjects when compared to healthy controls. Interestingly, a lower gas transfer and alveolar volume at rest, and their failure to increase during exercise in young children with previous BPD, suggested a reduced alveolar surface area in this population [39].

In contrast with these findings, Narang et al. [31] showed no exercise limitation in ex–preterm subjects when compared to term born controls. However, the preterm study group included few BPD subjects with also a relatively well-preserved lung function. In addition, other factors not related to lung function and diffusing capacity, such as deconditioning or perception of fatigue, could influence impaired exercise capacity. Landry et al. [40] reported that subjects with BPD were more sedentary than non-BPD and term subjects. Preterm subjects reported more frequently leg discomfort during exercise when compared to control subjects, reflecting not only deconditioning, but also an impaired peripheral muscle function related to prematurity [35,38].

Of note, several studies with adolescent and adult participants included individuals born during the pre-surfactant era, while studies with preschool and school-aged children referred to subjects born during the surfactant era. In addition, epidemiological studies investigating clinical outcomes used different definition and clinical evaluation of respiratory symptoms. Most of the studies used unstandardized questionnaires with unknown validity. The aforementioned methodological issues make difficult to compare studies and obtain a consistent interpretation.

## 3. Lifelong Lung Function in Ex-Preterm Subjects

In literature, follow-up studies of children and young adults born very-to-moderately preterm show persistent and significant lung function abnormalities. It is difficult to determine the role of prematurity in impaired lung function in childhood and adulthood because of the influence of aforementioned confounding factors. Furthermore, comparable study populations are few, as survival at extremely short gestations has been limited until recently [41,42].

### 3.1. The Relationship between Birth Weight and Lung Function

As originally hypothesized by Barker, low birth weight was implicated in poor lung function in adulthood [1]. Therefore, low birth weight children were initially investigated to assess the origin of lung impairment in childhood.

Already in the pre-surfactant era, Chan et al. [43] found that the long-term effect of prematurity depended greatly on low birthweight and hence prematurity itself, rather than neonatal respiratory treatment. The authors observed a reduced FEV1 in 130 children with a birthweight less than 2000 g at 7 years of age when compared to normal birth weight children, although FVC was preserved. Contrarily, McLeod et al. [21] found a reduced lung size expressed by a lower FVC in children aged 8–9 years with a very low birth weight when compared to controls. The authors also found that a reduced FVC was significantly associated with prolonged mechanical ventilation and the presence of pneumothorax.

After the introduction of surfactant therapy, Cazzato et al. [44] compared 48 very-low birth weight children to 46 age-matched term controls. The authors showed lower Z-score values of FVC, FEV1, FEF25-75, and higher residual volume and RV/total lung capacity (RV/TLC) ratio in ex-preterm children when compared to term controls at 8.5 years of age. These findings indicated an obstructive respiratory pattern associated with hyperinflation in very low birth weight children at school age.

Therefore, the association of low birth weight with lung function impairment was widely reported in literature [45,46,47]. A lower birth weight was associated with lower FEV1 in childhood [48] and adulthood [46,49] independently from premature birth, suggesting that lower birth weight led to a persistent reduction of airway patency. The association of birth weight with FVC was larger than the association of birth weight with FEV1, suggesting that lower birth weight may reduce lung function mainly in the airway capacity [45,48].

### 3.2. Lung Function in Infancy and Preschool Age

Several studies with participants at different ages were conducted to better understand the role of BPD and prematurity in lung function impairment across life phases. Studies in preschool children were few and showed increased interrupter resistance [50] and worse parameters at forced oscillation technique in ex-preterm children when compared to term born controls [51].

### 3.3. Lung Function in School Age and Adolescence

More studies have been carried out during school age. Verheggen et al. [27] found an impaired lung function with worse FVC, FEV1, reactance, and resistance in 118 ex-preterm children when compared to 32 term-born controls at 4–8 years of age. The authors stated that preterm children with and without BPD presented impaired lung function with airway obstruction, probably due to increased lung stiffness or peripheral lung abnormalities. The EXPRESS study (Extremely Preterm Infants in Sweden Study) found that preterm birth was associated with reduced maximal expiratory flows and lower lung volumes, measured by spirometry, and altered airway mechanics, measured by an impulse oscillometry technique at 6.5 years of age. Specifically, ex-preterm children showed a reduction of 9% of FVC and 13% of FEV1 with respect to term born controls [52]. Kaplan et al. [53] found similar FVC values in 28 ex-preterm children with BPD, 25 ex-preterm children without BPD, and 23 control subjects at 10 years of age. On the other hand, FEV 1, FEV0.75, and FEF25-75 values were lower in ex-preterm children, independently of the presence of BPD. Similarly, Simpson et al. [54] found lower FEV1, FEF25-75, and FEV1/FVC values in 163 ex-preterm children when compared to 58 term-born controls at 9–11 years of age, although the lowest values were in ex-preterm-children with BPD. In addition, the authors showed an association of a decreased lung function with lower gestational age and birth weight. In the EPICure study, Z-scores values of FEV1 and FEF25-75 were significantly reduced in 182 ex-preterm children when compared to 161 classroom controls at 11 years of age, mostly in the presence of BPD [24]. Filippone et al. [55] found an association between airflow obstruction severity at 2 years and spirometric alterations at 8 years in a group of moderate to severe BPD children. The authors speculated that airflow limitation during infancy was caused by early remodeling, and recovery was incomplete until childhood in children more severely affected.

### 3.4. Lung Function in Adulthood

Less studies evaluated lung function in adults who had survived BPD and premature birth. Gough et al. [34] reported that 72 BPD survivors had lower FEV1 and FEF25-75 when compared to 57 non-BPD preterm adults and 78 term-born controls at 24–25 years of age. Caskey et al. [35] documented reduced FEV1 and forced expiratory flow values in ex-preterm adults when compared to term-born ones, although the significance was achieved only for ex-preterms with BPD. Fixed airflow obstruction was more frequent in BPD survivors (25%) when compared to no-BPD preterms (12.5%) and term controls (0%). A recent longitudinal study demonstrated that BPD survivors with an airway obstruction early in life failed to achieve the expected optimal peak lung function at 24 years of age. In addition, the lung function measurement at different time points revealed a significant correlation between compliance of the respiratory system in the first days of life and z-score of maximal forced expiratory flow at functional residual capacity (zVmaxFRC) at 2 years, and between zVmaxFRC at 2 years and zFEV1 and zFEF25–75% at 15, 20, and 24 years [56]. Lastly, two recent meta-analyses, including 1.421 and 24.938 children, concluded that preterm birth negatively affected lung function and this lung impairment persisted into adulthood [10,48].

### 3.5. Gas Diffusion Impairment

Several studies assessed an impaired diffusing lung capacity (DLCO) in subjects born prematurely when compared to term born one [25,44,57,58]. Satrell et al. [57] found a reduction of DLCO of 10% in subjects born preterm when compared to term born controls, both in prepubertal and adolescent age. In the EPICure Study, 38 children born extremely preterm showed lower gas transfer values when compared to 38 term controls at 11 years of age [25]. Similarly, lower diffusing lung capacity values were found in 49 extremely preterm children when compared to classmate controls at 11 years of age [58].

An impaired gas diffusion in ex-preterm children was observed at 7–11 years of age [8,44,59] and in young adults of 19–20 years of age [60] and 24 years of age [35], independently of the presence of BPD. In our previous study, we also confirmed that prematurity affected gas transfer finding a positive association between DLCO values and gestational age in ex-preterm children with and without a prior diagnosis of BPD; this association persisted after adjusting for birth weight, CPAP duration, mechanical ventilation duration, breastfeeding, BMI, and sex [8].

An impaired acinar development characterized by fewer and larger alveoli, thickened alveolar–capillary membranes, and altered pulmonary capillaries induced a reduced pulmonary gas diffusing capacity in preterm infants [61]. Recently it was suggested that the alteration of diffusing capacity in ex-preterm children could be the hallmark of an underlying peripheral airway impairment at 10 years of age. Therefore, in school-aged children, prematurity effects were not only seen in airway obstruction, measured by spirometry, but also reduced capacity for gas exchange due to reduced alveolar and pulmonary capillary total surface area, detected by CO and NO diffusing capacity [62].

Although the real clinical significance of these changes remains unclear, periodic assessment of lung function is necessary in ex-preterm children from birth to adulthood, regardless of BPD. The correlation between childhood and adulthood lung function trajectories [63,64] suggests that ex-preterm children have an increased likelihood of COPD in later life. In addition, in order to better characterize the lung function in ex-preterm children, as well as lung volumes, diffusing capacity should be performed in association with spirometry.

### 3.6. Focus on Lung Function Impairment Related to Prematurity

Currently, there is increasing evidence that prematurity may influence lung function later in life, but it is still unclear whether it might lead to a respiratory restrictive or obstructive pattern.

Most of the studies showed a significant airflow obstruction with mean FEV1 values between 70 and 80% of the predicted values in children born preterm [14,18,36,62]. The strong positive association of gestational age with FEV1/FVC and FEF25–75% suggested that prematurity mainly affects airway development with an increasing susceptibility to develop obstructive lung diseases [13,45,48,54,65]. However, few studies showed a respiratory restrictive pattern in the first years of life [66,67]. We might hypothesize that typical lung pattern after BPD or very preterm birth is characterized by a combined restrictive and obstructive pattern that changes over time: restriction is more evident in very early life and obstruction later [18,66,67,68]. Reduced forced expiratory flows and volumes with stable FVC values might result from a reduced airway caliber due to chronic airway inflammation, airway remodeling, and/or reduced parenchymal tethering [15,54].

### 3.7. Focus on Lung Function Impairment Related to Bronchopulmonary Dysplasia

In literature, the independent effect of BPD and premature birth on lung function impairment is still debated. Two dated studies showed that preterm birth, as well as BPD, independently caused airway obstruction in childhood [39,69]. A recent finding showed similar lung function parameters in ex-preterm subjects with mild BPD when compared to ones without BPD in preschool age. The authors concluded that mild BPD might not lead to long-term respiratory consequences, independent of the effects of preterm birth [70]. A recent meta-analysis including 59 follow-up studies of ex-preterm subjects born between 1964 and 2000 showed that FEV1 was decreased in ex-preterm children, independently of BPD. Preterm-born groups without BPD presented a mean percentage of FEV1 that had reduced by 7.2% when compared to term-born controls. A larger difference was found between a preterm group with BPD and a term-born group; specifically, the preterm group with BPD, defined as supplemental oxygen dependency at 28 days and at 36 weeks postmenstrual age, showed a mean percentage of FEV1 reduction by 16.2 and 18.9%, respectively [10].

Recent evidence shows that lung function alterations are mostly related to gestational age rather than BPD [53,71]. However, much evidence still suggests that subjects with previous BPD show greater abnormality in lung function when compared to subjects without BPD [72,73].

Regarding sex differences, several studies found an increased vulnerability of boys than girls with similar gestational age and birth weight to developing BPD and reduced lung function during infancy and early childhood [74,75]. In contrast to these findings, Fawke et al. [24] found no sex differences in lung function and respiratory morbidity at 11 years of age. The male disadvantage in lung function could decrease over time, with girls tending to be at more risk of developing respiratory illnesses (i.e., asthma) post-pubertal maturation [24].

### 3.8. Focus on Lung Function Improvement over Time

In the last few years, evidence of improvement in lung function over time was reported in longitudinal studies [72,74]. A recent study assessing lung function at different time points (6, 12, 18, and 24 months) found that lung function improved gradually in preterm infants with mild to moderate BPD [74]. Thunqvist et al. [72] carried out a longitudinal follow-up study at 6 and 18 months of life of 55 infants born preterm with mild or moderate/severe BPD. The authors observed that all lung function parameters were below normal values in all subjects. Compliance of the respiratory system (Crs) values were normalized on average at 18 months of age, although significant differences in Crs persisted between groups with different BPD severity; at 2 years of age, the normalization of Crs might be due to an increased alveolarization. BPD severity did not predict lung function deterioration, but it could be related to an impaired alveolarization, as indicated by low Crs values.

Therefore, recent data suggested that in ex-preterm children, halted alveolarization could catch-up throughout childhood [75] and impaired lung function could improve over time, mostly in subjects without BPD [73,76]. These encouraging data would support the possibility of anatomical and functional recovery for lungs of ex-preterm infants with only marginal deficits of DLCO and exercise capacity [75]. Interestingly, lung function of BPD survivors has improved over recent years with the development of advanced therapy [10], giving hope for further improvement of lung function in future generations of ex-premature individuals.

Narang et al. [6] hypothesized that the apparent normalization of lung function over time could reflect a decreased sensitivity of spirometry instead of a real ‘‘catch-up’’ of lung growth. The authors remarked that spirometry is insensitive to distal airway obstruction until late phases of disease, suggesting the need to use more sophisticated methods to better investigate lung ventilation and lung growth, such as the lung clearance index and lung diffusing capacity.

A summary of lifelong lung function impairment is provided in Figure 3.

## 4. Lifelong Lung Structural Abnormalities in Ex-Preterm Subjects

In the past decades, the pattern of chronic lung disease has changed enormously in ex-preterm infants, mirroring a better perinatal management and an increasing lung immaturity with the survival of increasingly premature infants [77].

The information on structural lung abnormalities has been derived by autopsy specimens of ex-preterm children with severe BPD. Recently, computed tomography (CT) investigated lung structural abnormalities even in ex-preterms with less severe BDP, leading to new insights in pathophysiological mechanisms of BPD [12].

### 4.1. Lung Structural Abnormalities in Infancy and Preschool Age

In infants with BPD, many CT findings were similar to those observed in the pre-surfactant era and were still associated with supplemental oxygen and mechanical ventilation duration, despite the advances in neonatal care [78].

Few studies were carried out in the first years of life [79,80]. In a study involving 41 ex-preterm children between 10 and 20 months of life with new BPD, the authors found multifocal hyperlucent areas, linear opacities, and subpleural opacities; no bronchial involvement, such as bronchial wall thickening, was observed [79], compared to a previous study [81]. The hyperlucent areas reflect the abnormal alveolarization and distal vascularization. Linear and subpleural opacities were related to neonatal oxygen and mechanical ventilation exposure. However, the authors stated that oxygen was not necessarily the inducer of the lesions, but could be a marker of more severe lung disease [79]. The number of linear and subpleural opacities were significantly associated to low functional residual capacity, suggesting persistent fibrotic pulmonary lesions, and/or to a decrease of absolute lung volume due to halted septation [79]. In another study, areas of hyper-expansion and hyper-lucency were found both on chest radiograph and CT scans in preschool children with a history of BPD at 4 years of age [80].

### 4.2. Lung Structural Abnormalities in School Age

Simpson et al. [54] found that almost half of ex-preterm children during mid-childhood (9–11 years of age) showed bronchial wall thickening on chest CT reflecting post-inflammatory changes and/or ongoing airway inflammation. Importantly, children who had these anomalies on chest CT showed a worse obstructive lung disease and more respiratory symptoms when compared to their peers.

CT scans performed on 21 school-aged children (mean age of 12.7 years) with a history of new BPD showed at least one abnormality in 17 children (81%): linear-to-triangular subpleural opacities (71%), air trapping (29%), mosaic perfusion (24%), peribronchial thickening (14%), and emphysema (14%). The authors also found that CT abnormalities were associated with mechanical ventilation duration, BPD severity, and lower FEV1 values [82]. Therefore, structural lung abnormalities are common among school-aged children with a history of new BPD, resembling abnormalities described in the pre-surfactant era.

### 4.3. Lung Structural Abnormalities in Adolescence and Adulthood

The knowledge about CT findings of ex-preterm subjects is limited especially to adolescence and adulthood. Abnormalities were found in 87.5% of 72 children (10–19 years of age) born before 28 weeks of gestation in 1982–1985 (*n* = 40) and in 1991–1992 (*n* = 32): linear (80.6%) and triangular (58.5%) opacities, air trapping (26.4%), and mosaic perfusion (13.8%) [83].

Caskey et al. [35] found that young adult BPD survivors had radiological evidence of more severe structural lung impairment than non-BPD controls; the most common findings identified were subpleural opacities in 96% of subjects with BPD, compared to 43% of non-BPD subjects. These anomalies could derive from neonatal therapeutic insults that led to lung fibrotic changes. The authors also showed hypoattenuation on expiration (gas trapping) and bullous disease more frequently in subjects with BPD than in non-BPD ones (65 vs. 30% and 22 vs. 0%, respectively). There were no significant differences in emphysema, bronchiectasis, or bronchial wall thickening between two groups.

A recent systematic review of 16 studies confirmed structural abnormalities in more than 85% of chest CT in infants, children, and adults with previous BPD: decreased pulmonary attenuation, opacities, bronchial wall thickening, and consolidations. In addition, these lung abnormalities are often correlated to lung function deterioration and respiratory symptoms. At the end, the authors stated that none of used scoring systems were appropriately validated. For this reason, a standardized and validated chest CT quantitative scoring method for patients with BPD is needed to be defined [79].

A summary of lifelong structural abnormalities is provided in Figure 4 [84].

A summary table of symptoms, lung function alterations, and structural abnormalities is provided in Table 1.

### 4.4. Focus on the Role of Magnetic Resonance Imaging

As regards pulmonary magnetic resonance (MR), in premature infants with BPD, a significantly higher volume of “high signal lung” (i.e., signal over 45% of the patient’s mean chest wall signal with similar muscle mass/fat composition between groups) was described when compared to premature infants without BPD and healthy term infants [85]. There are also reports of higher lung T2 relaxation times in preterm infants with BPD. This finding may indicate increased interstitial remodeling as well as fibrosis, potentially associated with pulmonary inflammation and interstitial oedema. Similar findings were observed in adult patients with lung disease. Adams et al. [86] first reported that preterm infants had higher and more heterogeneously distributed proton density throughout the parenchyma than term controls. The authors also found that increased lung T2 relaxation time and decreased lung T1 relaxation time were associated with an overall increased risk for BPD, as well as an increased risk of more severe disease. In addition, in newborns with BPD, two types of parenchymal abnormalities were described: focal high-density areas and low-density, cyst-like abnormalities. [87,88] Furthermore, a cystic appearance of the parenchyma was only reported in BPD group and was absent in both preterms without BPD and full-term controls. This cystic appearance was probably due to emphysematous areas [88]. On the other hand, pulmonary MR showed that ex-preterm school-aged children with BPD had lower mean total proton density and lower proton density at full expiration when compared to those without BPD. These pulmonary MR findings were associated with greater residual volume and lung clearance index, suggesting the presence of air-trapping [89].

To date, newer and faster sequences and acceleration techniques have significantly improved the spatial resolution of pulmonary MR. However, lower resolution when compared to that of chest computed tomography and longer examination times limit the use of MR in pediatric pulmonary parenchymal imaging to the research field [85].

## 5. Follow-Up of Ex-Preterm Children

Most studies have largely focused on prevention rather than treatment of BPD. Although several tools have been studied for monitoring children with BPD, guidelines on comprehensive follow-up strategies for children with BPD are not clearly defined yet. Firstly, it is very important at discharge to plan the administration of palivizumab in children with BPD to avoid respiratory syncytial virus infection [90]. In addition, children with BPD should regularly receive scheduled vaccinations and annual vaccinations against influenza [91]. Daycare attendance for children with BPD should be evaluated with caution and on case-by-case basis [92].

The Italian Society of Infant Respiratory Disease recommends periodic pediatric respiratory examination in the first 3 years of life based on the severity of BDP. Lung function tests are recommended especially starting at 6 years of age (earlier if feasible) once a year [93]. Measurement of bronchodilation capacity and exhaled nitric oxide would be helpful. Reduced or absent response to the bronchodilation test and the normality of the exhaled nitric oxide fraction exclude bronchial hyperreactivity and eosinophilic inflammation as in asthmatic subjects [94]. In the first years of life, lung function tests are difficult to perform and radiological tests use ionizing radiation. Recently, lung ultrasound was proposed as a viable alternative for the monitoring of lung aeration and function in extremely preterm infants. In addition, gestational age-adjusted scores significantly predicted the occurrence of BPD, starting from the seventh day of life [95]. We proposed a clinical follow-up schedule after hospital discharge for children with BPD (Figure 5).

Recently, the European Respiratory Society (ERS) provided recommendations for the monitoring and treatment of children with BPD [92]. Regarding lung disease monitoring, lung imaging with ionizing radiation was recommended only in children with severe course of BPD, severe respiratory symptoms, and/or recurrent hospital admissions. Lung function evaluation was recommended to detect children at risk for lung and related vascular diseases in adulthood [92]. As suggested by ERS Task Force, we show the best lung function tests according to age in Table 1.

Regarding therapy, systemic or inhaled corticosteroids were not recommended by the ERS task force (low certainty of evidence), even if they could be used for children with severe BPD, severe respiratory symptoms, and recurrent hospitalizations, according to the treating physician. As regards bronchodilators, they could be taken into account in specific subgroups (children with severe course of BPD, severe respiratory or asthma-like symptoms, recurrent hospital admission due to respiratory morbidity, exercise intolerance, or reversibility in lung function). Both corticosteroids and bronchodilators should be monitored in a trial period before than being used for long periods.

Supplemental oxygen was suggested with a saturation target range of 90–95% [92].

Guidelines for children born prematurely without BPD are still lacking in literature. We recommend using the guidelines formulated for BPD even in premature infants without BPD, especially those with lower gestational age at birth.

## 6. Conclusions

With the recent improved survival of preterm newborns, it is mandatory to investigate the long-term effects of a premature birth in such a critical stage of lung development. Studies conducted so far often led to conflicting results. Previous studies often included cohorts born in the early 1990s, when surfactant use was not widespread, making findings not comparable to ones obtained from contemporary practice. The different gestational age of subjects included across the studies caused significant heterogeneity between. Additionally, the lack both of studies carried out in early childhood and of longitudinal studies make difficult to understand if the abnormalities reported in older ex-preterm children would have been evident earlier in life. Similarly, few studies investigated simultaneously respiratory symptoms, lung function, and structural alterations in ex-preterm subjects.

Recently, the respiratory consequences of prematurity are relatively well-described in literature, and an impairment of lung function is confirmed in childhood and adolescence. The effects of prematurity in adulthood have been less investigated and are therefore less clear. Several early-life events were linked to COPD in later life, and prematurity might facilitate its development. Therefore, long-term respiratory follow-up of preterm-born survivors is needed to detect lung function alterations in earlier stages and to establish a diagnosis of COPD as early as possible.

Although preterm birth should be not only of interest to pediatric pulmonologists, one study showed that this knowledge is not yet generally incorporated into daily practice when managing respiratory diseases [88]. Therefore, not only pediatricians but also pulmonologists and other physicians should be aware of this “new COPD of prematurity” for an accurate strategy of prevention. Lung-protective lifestyle interventions in ex-preterm subjects should be stressed in the primary care setting, including avoidance of smoking, vaccinations, vocational guidance, physical fitness programs, and weight control.

Lastly, in literature, the lung function decline, which begins in mid-adult life, might be more rapid or reach a critical threshold at an earlier age in those in whom maximum fetal and early childhood lung growth potential was not achieved. Therefore, it is important to identify those individuals born premature who are at greatest risk for developing long-term lung disease through a better awareness of physicians and a more comprehensive periodic lung function evaluation.

## Figures and Tables

**Figure 1 ijerph-19-05273-f001:**
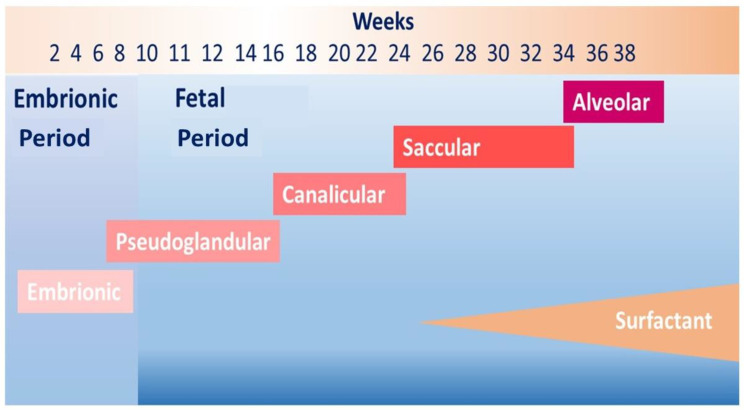
Lung development and preterm birth. The figure shows lung development phases from embryonic period to birth. In case of premature birth, in particular before than 30 weeks of gestation, the last generations of lung periphery and air–blood barrier are still forming.

**Figure 2 ijerph-19-05273-f002:**
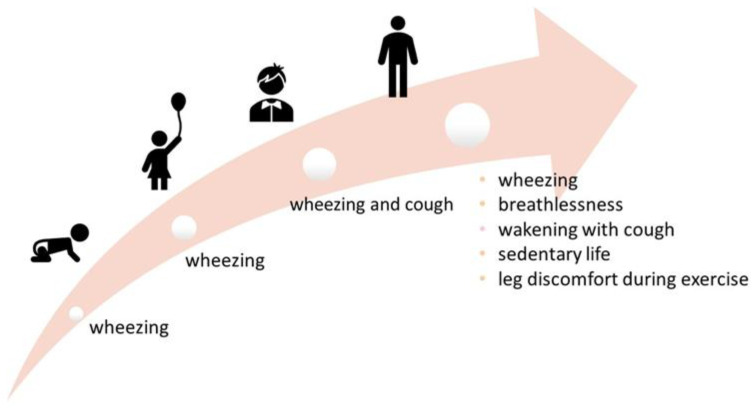
Respiratory symptoms at different ages. Most frequent symptoms at different ages are reported.

**Figure 3 ijerph-19-05273-f003:**
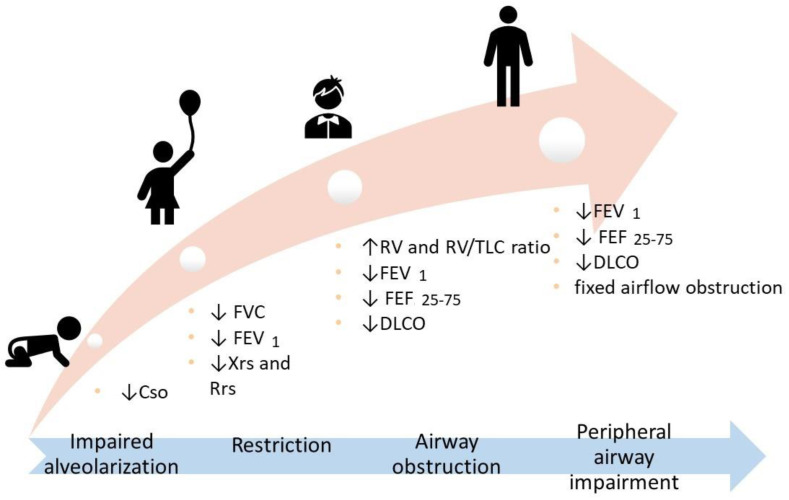
Respiratory function at different ages. Cso: Respiratory System Compliance; FVC: Forced Vital Capacity; FEV1: Forced Expiratory Volume in the 1st second; Xrs: Respiratory Reactance; Rrs: Respiratory Resistance; FEF25-75: Forced Expiratory Flow 25–75%; DLCO: Diffusion Lung Carbon Monoxide.

**Figure 4 ijerph-19-05273-f004:**
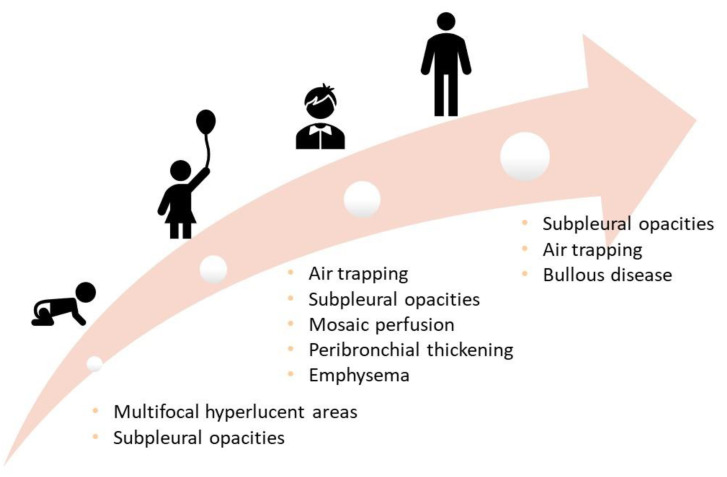
Structural abnormalities at different ages.

**Figure 5 ijerph-19-05273-f005:**
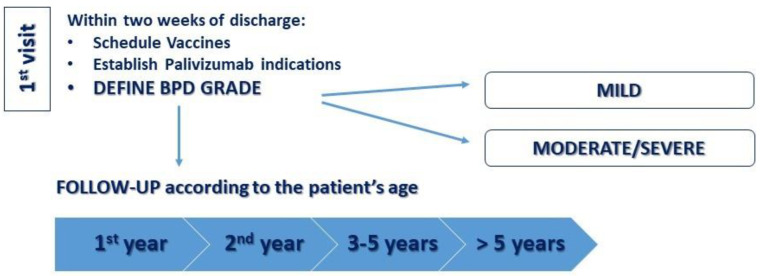
A proposed clinical follow-up schedule after hospital discharge. In this Figure, we propose our algorithm: in the first visit (within 2 weeks of discharge) BPD severity, vaccines schedule, and palivizumab indications are defined. During the first year of age, patients with mild BPD are evaluated by pediatric respiratory follow up visits at 3–6–12 months of life, while patients with moderate/severe BPD are evaluated at 1–3–6–9–12 months of life. During the second year of age, pediatric respiratory follow-up visits are performed every 3–6 months both in mild and moderate/severe BPD patients. Between the ages of 3 and 5, impulse oscillometry and resistance by interruption are performed annually or every 6 months. After 5 years of age, spirometry and diffusing capacity of the lung for carbon monoxide are performed annually or every 6 months.

**Table 1 ijerph-19-05273-t001:** Symptoms, lung function alterations, and structural abnormalities at different ages.

	Symptoms	Lung Function	Structural Abnormalities
**Toddler**	Wheezing	*Tidal flow-volume technique* *Rapid thoraco-abdominal compression technique (RVRTC)* ↓Respiratory System Compliance	*HRCT* Multifocal hyperlucent areas Subpleural opacities
**Preschool-age**	Wheezing	*Forced oscillation technique, multiple breath wash out tests, spirometry if compliant* ↓FVC, FEV1, reactance and resistance	*HRCT* Air trapping Subpleural opacities Mosaic perfusion Peribronchial thickening Emphysema
**School-age**	Wheezing Cough	*Spirometry and DLCO* ↑RV, RV/TLC ↓FEV1, FEF25-75, DLCO	
**Adolescence and adulthood**	Breathlessness Wakening with cough Sedentary life Leg discomfort during exercise	*Spirometry and DLCO, stress test* ↓FEV1, FEF25-75, DLCO Fixed airflow obstruction	*HRCT* Air trapping Subpleural opacities Bullous disease

## Data Availability

Not applicable.

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
