# Peer review of "Lifelong Lung Sequelae of Prematurity"

_ijerph, 2022, doi:10.3390/ijerph19095273_

Round 1

Reviewer 1 Report

This review article aims to describe the structural and functional sequelae due to preterm birth on the lungs.

Overall, the article is very interesting. However, it is quite difficult to follow the content which is very dense and requires sub-paragraphs which group the data either according to age or according to appearance during infancy and adulthood (the choice depends on authors but has to be followed throughout the paper). In addition, the key messages are difficult to grasp. They are generally proposed as almost all papers already noted. A new look helps the reader, especially a clinician, to improve not only his perspective vis-à-vis the pulmonary problems in ex-premature babies but also would help to ensure the most appropriate clinical follow-up by relying on the most recent knowledge. What are the propositions from authors? Are there guidelines already known by American or European pediatric lung associations that can be discussed here?

A reorganization of the content could make this review more attractive. For example.

  • Describe the stages of lung development in the human fetus, the newborn and during the 1st year of life to better follow the occurrence of the consequences according to the age of prematurity (a figure or diagram could be included)

  • Divide the sequelae according to age: during the 1st year of life, in preschool children, in young children, and then in the adulthood and adult (a table can clarify this issue. Indeed, discuss their occurrence according to gender and aggravating factors. For instance: The authors clearly describe the role of BPD, yet this consequence of pulmonary immaturity is more present in boys than in girls. Does this difference continue during the development of the child and in the appearance of pulmonary hypertension or COPD?

-   Adopt the same organization for the description of the information between the paragraphs

  • Give for each of these ages, the most practical test(s) for functional and structural lung exploration (figures are excellent and can be completed).

  • Propose key messages (for whom, when and how) the pulmonary follow-up of these children born prematurely would be likely to perform.

The recent papers by Crump C. et al., as well as the following ones are probably worthy of consideration in this review.

Vulnerability of the developing airway.

Ganguly A, Martin RJ. Respir Physiol Neurobiol. 2019 Dec;270:103263. doi: 10.1016/j.resp.2019.103263. Epub 2019 Aug 3. PMID: 31386914 Review.

Correlations between oxygen and positive pressure exposure in the neonatal intensive care unit and wheezing in preterm infants without bronchopulmonary dysplasia.

Glenn T, Ross KR, Trembath A, Tatsuoka C, Minich N, Hibbs AM. J Neonatal Perinatal Med. 2020;13(2):189-195. doi: 10.3233/NPM-190217.

Lung function in adults born preterm.

Näsänen-Gilmore P, Sipola-Leppänen M, Tikanmäki M, Matinolli HM, Eriksson JG, Järvelin MR, Vääräsmäki M, Hovi P, Kajantie E. PLoS One. 2018 Oct 19;13(10):e0205979. doi: 10.1371/journal.pone.0205979. eCollection 2018. PMID: 30339699

Lung consequences in adults born prematurely.

Bolton CE, Bush A, Hurst JR, Kotecha S, McGarvey L. Thorax. 2015 Jun;70(6):574-80. doi: 10.1136/thoraxjnl-2014-206590. Epub 2015 Mar 30. PMID: 25825005 Free article. Review.

Early Pulmonary Vascular Disease in Young Adults Born Preterm.

Goss KN, Beshish AG, Barton GP, Haraldsdottir K, Levin TS, Tetri LH, Battiola TJ, Mulchrone AM, Pegelow DF, Palta M, Lamers LJ, Watson AM, Chesler NC, Eldridge MW. Am J Respir Crit Care Med. 2018 Dec 15;198(12):1549-1558. doi: 10.1164/rccm.201710-2016OC. PMID: 29944842

Lung function and exercise capacity in prematurely born young people.

Harris C, Lunt A, Bisquera A, Peacock J, Greenough A. Pediatr Pulmonol. 2020 Sep;55(9):2289-2295. doi: 10.1002/ppul.24918. Epub 2020 Jun 29. PMID: 32568429 Clinical Trial.

Pulmonary function in former very low birth weight preterm infants in the first year of life.

Gonçalves DMM, Wandalsen GF, Scavacini AS, Lanza FC, Goulart AL, Solé D, Dos Santos AMN. Respir Med. 2018 Mar;136:83-87. doi: 10.1016/j.rmed.2018.02.004. Epub 2018 Feb 8. PMID: 29501252

Reduced Lung Function at Preschool Age in Survivors of Very Low Birth Weight Preterm Infants.

Chang HY, Chang JH, Chi H, Hsu CH, Lin CY, Jim WT, Peng CC. Front Pediatr. 2020 Sep 22;8:577673. doi: 10.3389/fped.2020.577673. eCollection 2020.

Development of lung function in very low birth weight infants with or without bronchopulmonary dysplasia: longitudinal assessment during the first 15 months of corrected age.

Schmalisch G, Wilitzki S, Roehr CC, Proquitté H, Bührer C. BMC Pediatr. 2012 Mar 23;12:37. doi: 10.1186/1471-2431-12-37.

Author Response

We thank the reviewer for his/her suggestions.

  • As suggested, we divided the manuscript in sub-paragraphs according to the age in order to facilitate the reader.
  • We discussed in the paragraph 5 entitled “Follow-up of children born prematurely” the follow-up for premature children proposed recently by the European Respiratory Society and the Italian Society of Pediatric Respiratory Diseases. Furthermore, we created a table (table 2) with follow-up proposed by the Italian Society of Pediatric Respiratory Society. We hope that key messages are now easier to grasp.
  • We created a figure (figure 1) illustrating lung development to better understand the occurrence of the consequences of premature birth.
  • We introduced several sentences on sex differences in the paragraph 3.6 “Focus on lung function impairment related to bronchopulmonary dysplasia”.
  • We described the most practical test(s) for functional and structural lung investigation according to ages in table 1.

Reviewer 2 Report

This is a logical and quite thorough review of an important subject. There are many minor grammatical errors that need to be fixed, and there are a number of more major issues. I really like the figures – they’re all very helpful.

MAJOR COMMENTS

Page 4, Section 3: I don’t think you can say these subjects show significant lung function deterioration. You’d need longitudinal studies to determine this. I’d replace this statement with: show persistent and significant lung function abnormalities.

Page 5 Line 190: I’m not sure the statement that the association of birth weight with FVC suggests that birth weight reduces airway function is true. FVC typically reflects lung size, more than airway function.

Page 6 line 256: Similarly, I don’t think reductions in CO and NO diffusing capacity reflects peripheral airway impairment – it’s much more likely to reflect reduced alveolar and pulmonary capillary total surface area and capacity for gas exchange.

End of Page 9: I’d suggest authors also mention new MRI modalities for assessing lung structure and function in BPD survivors – see Katz SL, Pulmonary Magnetic Resonance Imaging of Ex-preterm Children with/without Bronchopulmonary Dysplasia. Ann Am Thorac Soc . 2022 Jan 14. doi: 10.1513/AnnalsATS.202106-691OC.

Page 10 Line 413: this “new COPD of prematurity.” I think this is the area that needs more elaboration throughout the paper. COPD is pretty rare in young adults (unless they have alpha-1 AT deficiency) but asthma is common in children and adults. I think the authors need to describe the frequency of physician-diagnosed asthma in children and adults with BPD, ex-prems without BPD, and healthy controls, and also need to indicate the frequency of airway reactivity (bronchodilator response and/or airway challenge) in these populations. This is particularly important given the severe inflammatory insult that prematurity and ventilation represents. A brief description of inflammation in the pathogenesis of BPD would also be appropriate. Importantly, the definition of COPD is abnormal airway function (spirometry) after bronchodilator, so knowing the prevalence of abnormal FEV1 or FEV1/FVC post-bronchodilator would be important for defining the presence of early COPD.

MINOR COMMENTS

Page 2 Line 46: BPD is the most common form of CLD of infancy caused by treatment for respiratory distress syndrome in preterm infants [6]. CHANGE TO:  BPD is the most common form of CLD of infancy associated with premature birth and treatment for respiratory distress syndrome in preterm infants [6].

Line 58: and a less evidence of emphysema CHANGE TO: and less evidence of emphysema

Line 60: ADD: oxygen dependency after 28 days from birth AND radiographic changes and established…

Line 85: CHANGE SECTION TITLE TO: Lifelong symptoms in ex-preterm subjects

Page 3 Line 118: complained respiratory symptoms CHANGE TO: complained of respiratory symptoms

Line 131: To date, the balance of evidence in adolescence and adulthood suggested that ex pre-term subjects, independently of presence of BPD, have more respiratory symptoms, such as coughing, wheezing, and asthma [14]. CHANGE TO:  To date, the balance of evidence in adolescence and adulthood suggested that ex pre-term subjects, independent of the presence of BPD, have more respiratory symptoms, such as coughing, wheezing, and asthma, although some studies suggest symptoms may be more frequent in those with BPD [14].

Line 136: a shorted distance traveled on the treadmill [23,32], a greater CHANGE TO: less distance traveled on the treadmill [23,32], greater

Line 143: However preterm study group CHANGE TO: However the preterm study group

Page 4 line 146: impaired exercise capacity in ex-preterm subjects CHANGE TO: impaired exercise capacity [rationale: these factors can be important in anyone]

Line 151: Noteworthy, several CHANGE TO: Of note, several

Line 159: obtain a unique interpretation CHANGE TO: obtain a consistent interpretation

Line 172: already in pre-surfactant era CHANGE TO: already in the pre-surfactant era

Page 5 Line 210: independently of the presence of BPD. CHANGE TO: independent of the presence of BPD.

Line 217: mostly in presence of BPD CHANGE TO: mostly in the presence of BPD

Line 222: adults survived to BPD CHANGE TO: adult survivors of BPD

Page 6 line 226: although the significance was achieved CHANGE TO: although significance was achieved

Line 240: both in pre-pubertal and adolescent age CHANGE TO: both in pre-pubertal and adolescent age groups.

Line 253: indirect a reduced pulmonary CHANGE TO: resulted in a reduced pulmonary

Line 264: diffusing capacity should be performed in association to spirometry. CHANGE TO: lung volumes, diffusing capacity should be performed in association with spirometry

Page 7 Line 284: BPD, caused independently airway obstruction CHANGE TO: BPD, independently caused airway obstruction

Line 286: mild BPD might not lead to long-term respiratory consequences CHANGE TO: mild BPD might not lead to long-term respiratory consequences independent of the effects of preterm birth.

Line 296: much evidence still states that CHANGE TO: much evidence still suggests that

Line 317: in these subjects CHANGE TO: in future generations of ex-premature individuals.

Page 8 line 321: suggesting the use of more sophisticated CHANGE TO: suggesting the need to use more sophisticated

Page 9 line 343: no bronchial involvement, as bronchial CHANGE TO: no bronchial involvement, such as bronchial

Line 346: opacities are related CHANGE TO: opacities were related

Line 368: Lung abnormalities were found CHANGE TO: Abnormalities were found

Line 372: had a radiological evidence CHANGE TO: had radiological evidence

Line 383: lung function deterioration I THINK AUTHORS MEAN: lung function abnormalities

Line 384: none of used scoring CHANGE TO: none of the used scoring

Page 10 Line 419: early childhood growth potential CHANGE TO: early childhood lung growth potential

Author Response

We thank the reviewer for his/her suggestions.

  • As suggested, we modified the statement concerning lung function alterations in ex preterm subjects (line 203) and the one about reduction in CO and NO diffusing capacity (lines 305-306).
  • As regards FVC, as you say FVC reflects lung size, we used the expression “airway capacity”, meaning airways size rather than function (line 233).
  • As required, we provided a short summary of new application of chest MR in this setting in the paragraph 5 entitled “Focus on Magnetic Resonance role”
  • Concerning “new COPD”, we thank the referee for his/her comments. It is very tricky to establish the frequency of physician diagnosed asthma in ex-preterm children with or without BPD given that studies in literature are very heterogeneous regarding the sample size, time-point at which the respiratory outcomes were measured and gestational age of the participants. Few studies investigated the association of gestational age and BPD with post-FEV1 in childhood or adolescence. However, we would highlight that the introduction of the term “new COPD of prematurity” was mostly referred to the effect of perinatal factors, in particular the prematurity, on the development of COPD in adulthood, in contrast to old term of COPD which was mostly related to smoking, male sex or lower education level ( Soriano JB, et al. Arch Bronconeumol (Engl Ed) 2021 Jan;57(1):61-69)
  • Lastly, we fixed all the secondary issues that he/she highlighted.

Reviewer 3 Report

I thought this was a well-written and useful summary of lung disease of prematurity and will be interesting to paediatricians and respiratory physicians. I have a few minor corrections that should be addressed:

  1. Introduction, lines 68-72. This section is not referenced well and contains statements without a clear reference.
  2. Figures: please write a clear figure legend for each figure. The reader should be able to look at the figures and understand clearly what they mean without reading the paper.
  3. 3.1 The relationship between birth weight and lung function, line 169. Please reference.
  4. 3.4 Lung function improvement over time. Please explain Crs values.

Author Response

We thank the reviewer for his/her suggestions.

We provided an easy-reading legend for each figure and we also supplied the references required (lines 80, 210, 387). Crs values improvement is explained in lines 373-376; we particularly focused on lung function impairment because several studies reported that although pulmonary mechanics in BPD survivors improves during the first years of life reaching the range of normal values (Crs evaluation due to catch up growth), later most of these patients present abnormal airway function (paragraph 3 and particularly 3.3) . 

Reviewer 4 Report

Thank you for the possibility to review the mnuascript titled: “Lifelong Lung Sequelae of Prematurity”. The study is interesting and provides a valid analysis of the available data about pulmonary diseases in patients who were premature at birth. The quality of the review is high and includes most of the important punlication on the subject. There are a few minor comments:

-I would include a sum up table that includes age period, clinical signs that can be encountered, possible conditions, diagnostic modality (CT, spirometry, sputum culture etc) and management.

-Some of the diseases in early life that affect the lungs can cause conditions like Swyer-James syndrome, Mounier-Kuhn disease etc. Although these conditions are rare I would recommend to include a brief overview in this review as there may be a possible link.

-Conclusions should not contain citations and only a short summary with conclusions of the author

Please take into account the recommendations in the spirit of improving the quality of the submission.

Author Response

We thank the reviewer for his/her suggestions and comments.

  • As suggested, we provided a table with an overview of clinical signs, lung function tests and radiological findings (Table 1).
  • Concerning management of BPD, we introduced the management and therapy of BPD in the paragraph 5 entitled “Follow-up of children born prematurely”. We provided a summary of European Society guidelines for BPD about the use of steroids. In addition, we summarized the follow-up proposed by the Italian Society of Pediatric Respiratory Diseases in table 2. In the same paragraph, we reported the use of bronchodilation test in ex preterm children to differentiate respiratory symptoms of asthma from those related to BPD.
  • As suggested, we mentioned Mounier-Kuhn disease and Swyer-James syndrome because of their relation with prematurity (Lines 107-115).
  • As suggested, we also modified the conclusions.

Round 2

Reviewer 2 Report

Most of the major issues have been corrected, but Authors have introduced several new errors. I would strongly encourage to have revised version proof-read by someone whose first language is English before re-submission. 

MAJOR ISSUES:
Page 14 Line 500: Very few PFT labs outside research centers can do PFT's in children under 5-6 years of age. CHANGE: since 3 years of age TO: starting at 6 years of age (earlier if feasible) 

Line 517: As many studies have shown an increased prevalence of asthma (or asthma-like symptoms) in individuals born premature (especially if they have BPD), asthma therapy in accordance with international guidelines, including regular therapy with inhaled corticosteroids, are recommended in patients with an asthma-like phenotype, particularly in the presence of congruent pulmonary function changes. 

Table 21: Authors MUST obtain permission from original journal if this table has been scanned from another source. Table should be re-written, changing lung function evaluation to < 5-6 years and >5-6 years in the last row, and indicate table has been modified from the original source (give Reference). 

Page 16 line 558: I would not mention CT imaging scores for routine surveillance. Risk of ionizing radiation outweighs unproven benefits of early detection of lung disease, given lack of proven interventions at the present time. 

MINOR ISSUES

Page 2, line 46: CHANGE: since embrionic period to birth TO: from embryonic period to birth

Page 8 line 305: CHANGE: gas exchange due to alveolar TO: gas exchange due to reduced alveolar

Page 9 line 350: CHANGE show a greater lung function deterioration compared to TO: show greater abnormality in lung function compared to

Page 10 line 390: delete the last sentence (which doesn't actually make sense). Same with page 13, line 460.

Page 13 line 465: change title to: Focus on the Role of Magnetic Resonance Imaging
Line 467: please describe the significance of high signal lung. 
Line 471 change: oedema, likewise adult patients with lung disease TO: oedema. Similar findings were observed in adult patients with lung disease. 
Page 14 line 482 - explain the significance of low proton density. 
Line 496: CHANGE flu shots TO annual vaccinations against influenza. 

Page 15 line 534: CHANGE: caused a greater heterogeneity among TO: caused significant heterogeneity between 

Page 16 line 550 CHANGE: pulmonologist TO: pulmonologists
Line 552 CHANGE: smoking, vocational guidance TO: smoking, vaccinations, vocational guidance

Author Response

Thank to the Reviewer for His/Her suggestions. Concerning the major issues, we modified the starting age to perfrom pulmonary function tests (page 14 Line 500) and in agreement with the Reviewer decided to not suggest pulmonary CT as a surveillance tool, because of its risks (page 16 line 558).

About the therapy, we referred to the ERS task force (reference 92): bronchodilators and corticosteroids are not recommended in all patients, even if they could be used in selected patients according to the treating physicians with close monitoring (lines 521-528). 

As regards the table, we replaced it with a clinical follow-up schedule that could be proposed for children with BPD (figure 5). 

Funthermore, we modified all the minor issues as suggested by the Reviewer. Particularly, we explained the significance of high signal llung and low proton density (lines 467-68 and 483-487, respectively).